# Green Tea in Reproductive Cancers: Could Treatment Be as Simple?

**DOI:** 10.3390/cancers15030862

**Published:** 2023-01-30

**Authors:** Maclaine Parish, Gaelle Massoud, Dana Hazimeh, James Segars, Md Soriful Islam

**Affiliations:** Department of Gynecology and Obstetrics, Division of Reproductive Sciences & Women’s Health Research, Johns Hopkins Medicine, Baltimore, MD 21205, USA

**Keywords:** green tea, EGCG, antioxidant, pro-oxidant, reproductive cancers, ovarian cancer, cervical cancer, endometrial cancer

## Abstract

**Simple Summary:**

Green tea is a popular beverage worldwide and has shown to be beneficial in the treatment of different cancers. However, its role in the treatment of reproductive cancers remains controversial. This review aims to summarize the data available in the literature about the role of green tea in treating gynecological cancers. Examination of available evidence may provide a better understanding of the green tea benefits and focus future research related to this topic.

**Abstract:**

Green tea originates from the tea plant *Camellia sinensis* and is one of the most widely consumed beverages worldwide. Green tea polyphenols, commonly known as catechins, are the major bioactive ingredients and account for green tea’s unique health benefits. Epigallocatechin-3-gallate (EGCG), is the most potent catechin derivative and has been widely studied for its pro- and anti-oxidative effects. This review summarizes the chemical and chemopreventive properties of green tea in the context of female reproductive cancers. A comprehensive search of PubMed and Google Scholar up to December 2022 was conducted. All original and review articles related to green tea or EGCG, and gynecological cancers published in English were included. The findings of several in vitro, in vivo, and epidemiological studies examining the effect of green tea on reproductive cancers, including ovarian, cervical, endometrial, and vulvar cancers, are presented. Studies have shown that this compound targets specific receptors and intracellular signaling pathways involved in cancer pathogenesis. The potential benefits of using green tea in the treatment of reproductive cancers, alone or in conjunction with chemotherapeutic agents, are examined, shedding light on new therapeutic strategies for the management of female reproductive cancers.

## 1. Introduction

Green tea is one of the most popular non-alcoholic beverages consumed worldwide. It is composed of catechins, most prominently epigallocatechin-3-gallate (EGCG), epigallocatechin (EGC), epicatechin-3-gallate (ECG), and epicatechin (EC) [1]. Green tea, with its major compound EGCG, can have a variety of health benefits. However, its mechanism of action is complicated, as EGCG has been found to act as both a pro- and antioxidant in various cellular conditions. As an antioxidant, EGCG works to scavenge free radicals, chelate transition metals, activate various other antioxidants, and inhibit pro-oxidants [2,3]. This function is important for the regulation of downstream processes such as lipid peroxidation and cholesterol sequestration, and thus EGCG has also been shown to impact these pathways [4,5]. Conversely, many of the properties that suggest EGCG would be a potent antioxidant may contribute to pro-oxidant effects. This can be directly through the generation of free radicals as a byproduct of auto-oxidation or Fenton reactions, or through independent mechanisms [6,7]. These include functions such as driving apoptosis [8]. More recently, EGCG has been studied for its effects on epigenetic remodeling [9]. Taken together, there are many mechanisms by which EGCG can modulate cellular processes, many of which are relevant in disease contexts, particularly in cancer. EGCG has proven benefits in chemoprevention, as seen in studies of breast, prostate, kidney, colon, and liver cancer [10,11,12,13]. EGCG works through multiple processes to inhibit cell invasion and metastasis. It has also been shown that EGCG can mitigate the cancer driving effects of human herpesviruses infections [14]. These infections can drive a plethora of cancer phenotypes, including nasopharyngeal carcinoma, gastric cancer, squamous cell carcinoma, as well as several gynecological cancers subtypes [15,16,17]. EGCG’s role in gynecological cancer prevention and treatment remains controversial. This review summarizes the chemical and anti-cancer properties of green tea while detailing its molecular mechanism of action. We present the findings of several in vitro, in vivo, and epidemiological studies that examined the role of green tea in the prevention and treatment of reproductive cancers, including ovarian, cervical, endometrial, and vulvar cancers.

## 2. Methods

A comprehensive search of PubMed and Google Scholar up to December 2022 was conducted. The following keywords were used: green tea, EGCG, bioavailability, chemical composition, chemical structure, green tea metabolism, pharmacokinetics, pharmacodynamics, EGCG mechanisms, antioxidant, pro-oxidant, lipid peroxidation, apoptosis, epigenetic regulations, ovarian cancer, cervical cancer, endometrial cancer, vulvar cancer. The methodology of this study was that of a scoping review. This was conducted to identify current evidence of EGCG as a possible treatment or adjunct agent for gynecologic cancers and to identify and define possible knowledge gaps.

## 3. Green Tea: Chemistry and Bioavailability

Green tea is harvested from the tea plant *Camellia sinensis*. The fresh leaves of this plant are steamed, rolled, and dried, producing the famous green tea beverage. In contrast to white (slightly fermented), oolong (semi-fermented), and black (fermented) tea, green tea does not undergo fermentation [18]. The chemical composition of green tea has been well studied, as it combines a wide diversity of compounds (Figure 1). 

This includes alkaloids (caffeine, xanthine and theobromine), amino acids (notably L-theanine), carbohydrates such as sucrose, glucose and cellulose, lipids such as linoleic acid, vitamins (A, B2, B3, C, E and K), and pigments such as carotenoids and chlorophyll. Green tea also contains different trace elements, such as calcium, iron, copper, chromium, magnesium, and zinc, and is a particularly rich source of manganese [18,19]. Most importantly, up to 30% of the dry green tea leaf weight is composed of polyphenolic compounds known as catechins (flavan-3-ol), the standard green tea flavonoids. The four major green tea catechins are epigallocatechin-3-gallate (EGCG), epicatechin-3-gallate (ECG), epigallocatechin (EGC) and epicatechin (EC) (Figure 2) [20,21].

Catechins are best known for their potent antioxidant properties, which highly rely on the presence of multiple hydroxyl groups in their chemical structure [19,20,21,22,23,24,25]. A considerable number of existing studies in the literature have examined the antioxidant capacity of catechins in vitro and in vivo and compared them to other popular antioxidants. For example, green tea catechins were shown to be stronger antioxidants than vitamins C and E, whereby one cup of green tea was estimated to be equivalent in antioxidant power to 100–200 mg of pure ascorbic acid [22,23,25,26,27]. This is in accordance with the fact that these polyphenolic compounds can act as radical scavengers, as well as chelating agents for metal ions such as iron and copper [24,28,29]. Interestingly, antioxidant effectiveness varies among different catechin derivatives, and has been shown to positively correlate with the number of hydroxyl groups. EGCG, which has eight hydroxyl groups, is the major bioactive ingredient in green tea and is considered the most potent catechin derivative [21,25,29].

Green tea has an exceptionally high catechin content when compared to other tea types, and this can be attributed to the different techniques used in tea manufacture and preparation. Every type of tea comes from the same plant, but as mentioned above, green tea is the only type that remains unfermented. During the fermentation process, tea leaves undergo an auto-oxidation reaction catalyzed by the enzyme polyphenol oxidase. As a result, tea polyphenols (catechins) become oxidized and form polymers known as theaflavins and thearubigins. Theaflavins and thearubigins are the predominant compounds in fully fermented black tea. In contrast, catechins in unfermented green tea do not undergo oxidation, retaining their chemical structure and properties [19,30,31]. Studies comparing bioactive compounds in different teas suggest that green tea contains considerably higher catechin levels, especially of EGCG and EGC [19]. Given the positive correlation that was found between total phenolic content and antioxidant capacity in different assays, green tea is thought to exhibit the most potent antioxidant activity among other teas [19,20,21,25,30,31,32,33].

The effects of catechins have been well characterized in vitro; however, bioavailability and metabolism factors must be considered before establishing these effects in vivo. Data from different pharmacokinetic studies reveal that the half-life (*t1/2*) of EGCG, the major green tea catechin, is 5 ± 2 h, [34,35,36,37] and its peak plasma concentration is reached at approximately 2 h after ingestion [34,35,36,38,39]. Following green tea consumption, a minor fraction of intact catechins reaches the systemic circulation. The bioavailability of catechins is limited by factors such as metabolism and intestinal transport. Most catechins undergo extensive phase II metabolism in the small intestine and liver, while remaining excess catechins enter the colon and are degraded by resident microorganisms. Phase II conjugation processes include sulfation, glucuronidation (intestine and liver) and methylation (liver). The resulting methylated, sulfated, or glucuronide catechin metabolites are absorbed in the liver and small intestine. Through enterohepatic recirculation, some of these metabolites also pass into the colon where they are catabolized by the resident microflora into smaller phenolic acids and ring fission products that are reabsorbed into the circulation and excreted into urine [39,40,41,42,43]. Evidence suggests that not only catechins, but also their conjugated metabolites and microflora-mediated catabolites are absorbed and can contribute to the in vivo biological effects of green tea [34,39,40,41]. The absorption of catechins and their metabolites occurs in the intestine via efflux transporters such as multidrug resistance-associated protein 2 (MRP2) and P-glycoprotein (PgP). Therefore, the bioavailability of catechins is thought to be further influenced by their affinity to these transporters and the presence of certain drugs that interact with PgP and MRP2 [44,45,46,47]. The presence of food can sometimes interfere in the absorption of catechins, but the effect varies and is not very well understood. Some studies have found that carbohydrate and vitamin C consumption improves catechin bioavailability [48,49,50]. In contrast, milk was not shown to have a significant effect [37,51].

In an attempt to overcome the poor bioavailability of green tea EGCG, several groups have utilized synthetic EGCG derivatives, also known as EGCG prodrugs generated by acetylation of the reactive hydroxyl groups [52]. This was shown to protect the hydroxyl groups and enhance the stability, bioavailability and biological potency of EGCG [53]. Pro-EGCG analogs have demonstrated greater anti-oxidative and anti-angiogenic capacities than EGCG in mice endometrial implants as well as human leiomyoma cells [54,55]. Furthermore, Pro-EGCG exerted significantly stronger antitumor effects than EGCG in skin and breast cancer cell lines and experimental cancer mouse models [56,57,58,59,60]. These findings provide insight into the enhanced properties of Pro-EGCG analogs and their potential use for cancer prevention and treatment.

## 4. Green Tea: Mechanisms of Actions

### 4.1. EGCG-Interacting Proteins

EGCG interacts with many proteins. Here, EGCG binding interactions relevant to understanding its interaction with cancer phenotypes are described. Many of these interactions induce physical or conformational changes of the proteins resulting in altered activity. Many EGCG protein interactions have been studied via affinity gel chromatography (AGC). Human plasma proteins fibrinogen, fibronectin, and histidine-rich glycoprotein (HRG) were found to bind EGCG through AGC [61]. Many of these proteins are important in the formation of ECM, and thus are relevant in cancer metastasis and invasion mechanisms. Similarly, EGCG was found to bind several metalloproteinases, specifically matrix metalloproteinase-2 (MMP-2), and matrix metalloproteinase-2 (MMP-9), through AGC [62]. MMP family proteins are classified as oncogenes, functioning to degrade ECM components to drive tumor cell migration [63]. Through binding to MMPs, EGCG inhibited cell invasion [62]. These data have since been confirmed by gelatin zymography, with the gallate group serving an important role in facilitating this interaction [64]. Additionally, it was found that matrix metalloproteinase-2 (MMP-2) exhibits similar interactions with EGCG, with MMP-12 and MMP-9 exhibiting higher sensitivity to EGCG directed inhibition than MMP-2 [64]. AGC was also used to identify protein interactions with the intermediate filament, vimentin [65]. It was reported that EGCG inhibited the phosphorylation of several vimentin residues (Ser50 and Ser55), ultimately inhibiting vimentin activity [65]. AGC methods showed EGCG was able to stabilize the interaction between heat shock protein 90 (HSP90) and the Aryl hydrocarbon receptor (AhR) through direct binding interactions with HSP90 [66]. Co-immunoprecipitation experiments validated this conclusion, showing that EGCG binding stabilized the entirety of the AhR complex, inhibiting the downstream effects (such as AhR-mediated transcriptional events) because the chaperone HSP90 fails to dissociate [67]. Stabilization of the interaction keeps AhR in its inactive state, preventing it from binding ligands. AhR is commonly upregulated in cancer, and thus, through inhibiting its activation, EGCG may mitigate the effects it has on tumorigenesis. EGCG was found to directly bind in the ATP-binding site of glucose-regulated protein 78 (GRP78) to inhibit its activity through AGC [68]. GRP78 contributes to many cellular processes, such as: the unfolded protein response (UPR), proliferation pathways (PI3K/AKT), inhibition of fatty acid synthesis, and mitochondrial oxidation [69]. Thus, through the inhibition of GRP78 activity through binding interactions, EGCG can inhibit many of the pathways commonly involved in tumorigenesis. Similarly, EGCG was found to inhibit the nucleotide binding domain 2 (NBD2) of P-glycoprotein (P-gp), an efflux transporter pump, by occupying the ATP binding site [70]. This was identified via a molecular docking algorithm, Autodock, that generated 3D structures of EGCG and NBD2 to locate potential binding interactions [70]. Because P-gp is important in drug transport, it is commonly active in drug-resistant cancer phenotypes working to pump drugs out of the intercellular environment [70]. Through interactions with EGCG, P-gp-mediated drug resistance can be decreased, through preventing cellular export. AGC validated the binding interaction between EGCG and Fas to promote Fas activity in the cell [71]. Fas, a transmembrane death receptor protein, is important in inducing apoptosis. In this case, ligand binding activates Fas to trigger the caspase cascade responsible for driving apoptosis [72]. By promoting this activity, EGCG may play an important role in driving apoptosis in cells that have tumor driving mutations working to inhibit these processes. NMR analysis techniques were used to identify inhibitive EGCG binding interactions with B-cell lymphoma-extra large (Bcl-XL) and B-cell lymphoma-2 (Bcl-2), two antiapoptotic proteins [73]. Similar to its role in promoting Fas-mediated apoptosis, EGCG again acted to promote apoptosis. Thus, EGCG is capable of inhibiting two antiapoptotic proteins that are commonly upregulated in cancer to contribute to prevent cells from dying [74]. EGCG binding 67 kDa laminin receptor (67LR) has been well characterized with high affinity through plasmon resonance [75]. This receptor is important in metastasis, as it functions in cell adhesion to the basement membrane, and by binding EGCG, its activity is inhibited. Human peptidyl prolyl cis/trans isomerase (Pin-1), another oncogene, was found to bind EGCG through X-ray crystallography and pull-down assays [76]. EGCG binds the WW domain to prevent substrate binding, and thus inhibit the activity of components downstream of Pin-1, such as nuclear factor-kB (NF-kB) and cyclin D1 that are important in tumorigenesis [76]. In another AGC experiment, EGCG was found to bind transforming growth factor-beta (TGF-β) type II receptor (TGFR-II) and inhibit downstream activity of TGF-β pathways. TGF-β signaling regulates many critical cellular functions, such as differentiation, migration, apoptosis, and is commonly mutated in disease phenotypes. Among many other proteins identified to have binding interactions with EGCG, these are the most important for understanding the potential roles of EGCG in the context of cancer phenotypes.

### 4.2. Antioxidant Properties of EGCG

Antioxidants are broadly defined as compounds that function to decrease oxidation in cells through the regulation of free radicals [77]. In the absence of antioxidants, oxidative stress can be dangerous for cells because of its potential to inflict DNA damage [77]. This can ultimately contribute to many disease states, so it is important to prevent cells from remaining in a state of oxidative stress. EGCG has been found to act as an antioxidant through many distinct pathways (Figure 3). These antioxidant effects can be categorized into four main groups: direct scavenging of free radicals, chelation of transition metals, activation of other antioxidants, and inhibition of pro-oxidant compounds [2,3].

First, EGCG, along with other catechin compounds, works directly as a free radical scavenger to protect cells from oxidative stress [78]. In this way, EGCG acts as an antioxidant, working to reduce the cellular concentrations of reactive oxygen species (ROS) and other free radicals produced in times of oxidative stress that can be damaging to cells [78]. There are many structural components of catechin compounds that allow for direct interaction with ROS. The diphenyl propane group and saturated heterocyclic ring of catechins enhance their stability through electron delocalization that uniquely contributes to their ability to scavenge free radicals [78]. These phenolic structural motifs allow EGCG to act as a hydrogen donor to capture free radicals [79]. Unlike other catechins, EGCG also has a gallic acid structural motif that adds to its antioxidant properties [79]. Because of this gallate group, EGCG is able to act as a superior electron donor compared to other catechin compounds, increasing its functionality in scavenging free radicals [79]. While most of the literature focuses on EGCG’s role in the sequestration of ROS, it also functions to scavenge reactive nitrogen species (RNS), produced from a reaction between nitric oxide (NO) and superoxide. The mechanism by which EGCG inhibits RNS accumulation is not fully understood, although many of the structural antioxidant properties contributing to the scavenging of ROS may play a similar role here. Alternatively, this may be linked to other antioxidant properties of EGCG, such as the inhibition of pro-oxidant compounds important in the signaling cascade that produces NO [80].

In a second direct antioxidant mechanism, it has been observed that EGCG plays an important role in chelating transition metals. In this case, EGCG functions to sequester metal ions that play a causal role in the generation of free radicals through interactions with various peroxidases [6]. This is likely the result of another functional property of the phenolic groups of EGCG. Although this role in chelating transition metals is important, it is thought to be less consequential than free radical scavenging in characterizing antioxidative properties of EGCG [81].

Next, EGCG plays important roles in the regulation of other antioxidant compounds. Generally, ROS accumulation is influenced by mitochondrial function (as the majority of ROS are produced as a result of oxidative phosphorylation reactions occurring here) and activity of various players in antioxidative response. EGCG plays an important role in both processes [82]. EGCG has been found to contribute to the upregulation of antioxidant response genes such as superoxidase dismutase (SOD) and glutathione peroxidase (GPx) [2]. Similar to the functions of EGCG outlined above, these proteins work as important factors in the conversion of free radicals to stable molecules in order to prevent them from inflicting cellular damage. Mitochondria are particularly vulnerable to attack by ROS in times of oxidative stress because they are responsible for much of their production [82]. Thus, mitochondrial function serves as an indicator of cellular stress. Through another level of regulation, EGCG was shown to increase mitochondrial potential, a measure of efficiency of mitochondrial oxidative phosphorylation, and thus a measure of mitochondrial function [82].

Recently, it was shown that EGCG is also able to inhibit the activity of several pro-oxidant compounds. EGCG has been shown to inhibit the following pro-oxidant enzymes: NADPH oxidase (NOX), COX-2 and xanthine oxidase [3]. These enzymes all have functions in the generation of ROS in times of stress induced by different cellular conditions. NOX is a major source of ROS production, and is thus an important pro-oxidant enzyme [3]. Additionally, as previously mentioned, EGCG works to inhibit enzymes involved in the production of NO. EGCG has been shown to inhibit inducible NO synthetase (iNOS), the enzyme responsible for NO production [3]. This inhibits accumulation of RNS as RNS are produced from NO in times of oxidative stress [3]. Overall, EGCG works to inhibit pro-oxidant enzymes that are important for the generation precursors of both RNS and ROS.

### 4.3. ECCG and Lipid Peroxidation

Connected to the above discussion of properties of EGCG as an antioxidant, it has been observed that EGCG can inhibit lipid peroxidation. Lipid peroxidation is the process by which free radicals, such as ROS, oxidize lipids, creating dangerous byproducts that have been implicated in the progression of many serious physiological conditions [83]. For reasons described above, EGCG can sequester many of the cell’s free radicals, therefore reducing their interaction with lipids, and the peroxidation reaction that follows. Thiobarbituric Acid Reactive Substances, or TBARS, serve as an indicator of cellular levels of lipid peroxidation, as it is produced as a by-product of this reaction. In an in vitro experiment assessing the potency of various antioxidant compounds in the reduction of lipid peroxidation in conditions of oxidative stress induced by high concentrations of hydrogen peroxide or ferrous ammonium sulfate, EGCG significantly reduced TBARS accumulation, suggesting that EGCG acts as a potent inhibitor of lipid peroxidation [4]. As expected, EGCG was also found to be the most protective inhibitor from the green tea polyphenol family due to its structurally unique antioxidant qualities outlined above [84]. Not only was this activity attributed to EGCG’s ability to scavenge free radicals and chelate free metal ions, but it was found to also be influenced by the stability of their semiquinone, or free radical state [84]. When EGCG donates hydrogen molecules in the scavenging of free radicals, it is often left in its own free radical, or semiquinone state. Because of the gallate group of EGCG, its semiquinone structure is more stable than many other catechins, thus contributing to its effects on lipid peroxidation [78].

### 4.4. Hypocholesterolemic Properties of EGCG

EGCG may also influence the regulation of cholesterol levels in the blood. Cholesterol must be tightly regulated, as it is a critical component of the cell membrane and contributes to the biogenesis of steroid hormones. To begin, there have been many clinical trials conducted to directly test the effect of EGCG on the accumulation of serum lipids. Collectively, the results suggest that EGCG can significantly reduce total cholesterol and low-density lipoprotein (LDL) cholesterol levels in human patients [5].

The danger of LDL cholesterol arises when it builds up in the walls of arteries, contributing to the formation of atherosclerotic plaques. These plaques are dangerous not only because they can cause blockage in arteries, but also because LDL can be oxidized when it accumulates in this way. As detailed above, EGCG can act as an antioxidant compound, and thus can also act in these circumstances to prevent oxidative stress. In a human clinical trial, LDL oxidation decreased after treatment with EGCG, and further in vitro studies showed EGCG could incorporate itself into the LDL particle in order to physically inhibit its oxidation [85]. EGCG has also been found to function at other levels of LDL processing. The cell naturally attempts to prevent the formation of atherosclerotic plaques via removing LDL from circulation through binding to LDL receptors (LDLRs) in the liver. This binding interaction is facilitated by apolipoprotein-B (apoB). In an in vitro study in Heg2 cells, EGCG was found to inhibit apoB secretion [86]. This is important because apoB is necessary for LDL to circulate through the blood. By inhibiting apoB, EGCG sends LDL to be stored in cytosolic lipid droplets, through binding with LDLRs, rather than remaining in circulation [87]. By decreasing the circulating LDL, this is another mechanism by which EGCG prevents its oxidation.

### 4.5. Pro-Oxidant Properties of EGCG

Although less understood, many of the mechanisms that make EGCG a uniquely effective antioxidant compound may lead to pro-oxidant effects in certain cellular conditions. For example, via its role of chelating metal ions, EGCG may alternatively facilitate reduction reactions, generating free radicals from bound metals. For example, it has been found that when associated with iron, EGCG promotes the progression of Fenton reactions, reducing FeIII to FeII, producing hydroxyl radicals as a cytotoxic byproduct [6]. In this way, EGCG may have dual functions as both a pro-oxidant and an antioxidant.

There are also pro-oxidant functions of EGCG that are separate from the pathways affected when EGCG acts as an antioxidant. For example, EGCG can undergo auto-oxidation in physiological conditions to produce superoxide anion radicals as a byproduct [7]. These free radicals can further oxidize EGCG, ultimately resulting in the production of hydrogen peroxide [7]. This role in the generation of ROS such as hydrogen peroxide and superoxide makes EGCG act as a pro-oxidant, leading to conditions of cellular stress [7]. In this way, many of the structural properties that make EGCG uniquely capable of capturing free radicals also make it an effective producer of the same reactive oxygen species it was initially thought to protect cells from.

It is likely the balance between antioxidant and pro-oxidant effects is dictated by several factors described below. First, the effects of EGCG have been found to be dose dependent [7]. At low levels, EGCG was found to induce the antioxidant properties outlined above, while at higher doses it favors pro-oxidant effects [7]. In this way, at high doses, EGCG may contribute to the same stress pathways it was initially characterized to inhibit. Although generation of free radicals is generally hazardous to cellular function, there are circumstances in which a pro-oxidant can be beneficial. In this way, the complicated role of oxidation in the physiological environment is important to consider when understanding the potential applications of EGCG in clinical settings. For example, EGCG could be used as a preventative tool in cancer treatment at low doses to serve as an antioxidant in the prevention of DNA damage, and as a pro-oxidant later in cancer progression as a driver of apoptosis [7]. These effects, however, may not be clearly separate from one another. In situations where EGCG has been shown to function as a pro-oxidant, there was also an upregulation of various antioxidant pathways, suggesting that both functions can occur in conjunction with one another [88]. In summary, while the relationship between the pro-oxidant and antioxidant roles of EGCG is not fully characterized, it is important to understand the full scope of potential interactions and effected pathways when applying its use to clinical settings (Figure 4).

### 4.6. EGCG and Apoptosis

The above pro-oxidant functions of EGCG align with its properties as a driver of apoptosis. While some of this activity can be explained by the accumulation of ROS and driving cellular oxidative stress, there are other interactions unique to apoptotic pathways that EGCG can direct [8]. While EGCG has been widely studied for its role in inhibiting invasion and cell proliferation in human cancer models, its role in promoting apoptosis is less studied. There are a variety of signaling pathways and mechanisms by which EGCG mediates apoptosis. Regulation of apoptosis is crucial for cell turnover and proper development and is thus controlled by a wide array of cellular processes [89]. This mechanism of programed cell death is frequently found to function irregularly in human disease, most commonly cancer and neurodegeneration [89]. There are three broad pathways that can induce apoptosis through caspase-dependent mechanisms: extrinsic, intrinsic (cellular damage), and T-cell-mediated cytotoxicity [89]. The two broad pathways relevant to EGCG action are categorized as external (ligand-receptor-mediated) and internal (mitochondrial-driven) [90]. Importantly, it has been suggested that EGCG can specifically induce apoptosis in cancer cells, leaving healthy cells alone. In a study of hepatocellular carcinoma, EGCG specifically induced apoptosis in cancerous cells [90]. This was mediated primarily through the internal, or mitochondrial-mediated apoptotic pathway [90]. Here, it was hypothesized that EGCG treatment led to a downregulation of NF-κB, an important cell proliferation and cell survival signal [90]. By inhibiting NF-κB, downstream targets, such as Bcl-2 (binds and inhibits Bax) and Bcl-2-associated X protein (Bax) (proapoptotic factor) activity were impacted, driving mitochondrial membrane potential to decrease to activate the caspase cascade via the release of cytochrome c [90]. In this same model, it was shown that EGCG could increase the expression of p53 in order to arrest the cell in G1, another apoptotic signal [90]. This effect is thought to be mediated by a direct binding interaction between p53 and EGCG. It was recently reported that EGCG can bind the N-terminal domain of p53 in order to prevent ubiquitination by the E3 ligase, Mouse double minute 2 (MDM2) protein, thus preventing its degradation in vitro [91]. It is likely these effects were largely contained to cancerous cell types, as the protein targets, such as p53 and NF-κB, are frequently mutated in cancer cells. Similar results have been found in other malignant tumor cells, such as breast, pancreatic, adrenal, and gastric cell lines [92,93,94,95].

In separate studies, it has been shown that EGCG can also promote apoptosis via the external ligand binding pathway through the Fas cell death receptor. In Fas-mediated apoptosis, ligand binding activates the transmembrane receptor protein, Fas, to activate the caspase cascade that triggers apoptosis [72]. Broadly, the effect of EGCG is thought to be mediated by direct binding interactions between Fas and EGCG, first characterized in human monocytic leukemia U937 cells [71]. In adrenal cancer NCI-H295 cells, EGCG was shown to drive activity of caspase-8, the downstream target of Fas, and to promote Fas-mediated induction of this signaling cascade to drive apoptosis [95]. As with the internal pathway, these Fas-mediated effects of EGCG have been observed in many cancer types: head and neck squamous carcinoma cells (HNSCCs), pancreatic cancer cells (MIA PaCa-2), and human liver cancer (HepG2) [96,97,98]. Similar to the data above, EGCG induced apoptosis through this pathway specifically in cancer cells [99]. This unique targeting feature makes EGCG a promising therapeutic option in cancer treatment. Finally, it has been shown that EGCG can inhibit antiapoptotic proteins. Specifically, EGCG can bind glucose-regulating protein 78 (GRP78) to modulate its function in suppressing apoptosis [68]. GRP78 functions to inhibit cellular caspase cascades, thus suppressing apoptosis. This protein is uniquely upregulated in multiple drug-resistant cancers, and thus EGCG provides a new treatment option for such cancers where apoptosis is inhibited even when treatment is administered. Through direct binding interactions with GRP78, EGCG was shown to inhibit the ATPase activity of GRP78, thus inhibiting its downstream effects [68]. Additionally, EGCG can bind both Bcl-XL and Bcl-2 [73]. The role of these proteins has been described above. Through this binding interaction, Bcl-2 is no longer able to bind Bax, and thus Bax can induce apoptosis in the presence of EGCG [73].

### 4.7. EGCG’s Role in Epigenetic Regulation

Epigenetic modifications play a crucial role in driving cancer phenotypes, primarily through inactivation of tumor suppressor genes. One common mechanism is hypermethylation of CpG island promoter regions of tumor suppressor genes. This can drive cancer phenotypes by silencing the expression of genes that work in tumor suppressor pathways such as the regulation of cell proliferation, cell cycle, or apoptosis, among others. EGCG has been characterized as an epigenetic modulator in many cancer types through a variety of mechanisms. First, EGCG can influence epigenetic modifications through targeting epigenetic modulators, such as DNA methyltransferases (DNMTs) and histone deacetylases (HDACs). Studies in human cervical cancer cells showed that EGCG can directly bind DNMT1, DNMT3B and HDAC to inhibit their respective activities [9]. This inhibition results in epigenetic changes. For example, in a prostate cancer model of DUPRO and LNCaP cells, EGCG treatment decreased accumulation of H3K27me3 in the promoter region of the tissue inhibitor of the matrix metalloproteinases (TIMP-3) gene, a potent tumor-suppressor gene that binds MMPs to reduce invasion [100]. By increasing accessibility to the promoter region through inhibition of methylation, EGCG drives gene expression. Along with this, acetylation of H3K9/18 was increased, likely due to a decrease in HDAC activity, suggesting further activating epigenetic modification facilitated by EGCG [100]. It was also shown that HDAC and DNMT enzymatic activity was significantly decreased after EGCG treatment, suggesting this was the mechanism of action in effecting acetylation and methylation status [100]. Similar conclusions have been drawn in other cancer models. For example, in a model of acute promyelocytic leukemia (APL), EGCG similarly decreased levels of H3K9me2 on genes important for cell cycle arrest and differentiation [101]. These effects were likely a result of the downregulation DMNT1 [101]. Similarly, the investigators also observed a reduction in various HDACs and an increase in acetylation in these genes [101]. Together, these data show that EGCG can function to drive expression of tumor suppressor genes that are typically inhibited in cancer through the regulation of epigenetic remodelers.

Separately from methylation and acetylation events, EGCG may have a role in the regulation of microRNAs, small regulatory RNAs responsible for gene silencing events. MiRNAs function by binding to the 3′UTRs of mRNAs with sequence complementarity to inhibit their translation. Here, they silence gene expression by triggering mRNA decay pathways. It has been found that EGCG can lead to the transcriptional silencing of miRNA-encoding regions, allowing their downstream targets to be transcribed. While it is not entirely understood which miRNAs EGCG can specifically target, there are data suggesting EGCG targets miRNAs important for the regulation of AKT, MAP kinases and cell cycle regulation. These pathways are important for cell proliferation and differentiation and are thus commonly mutated in cancer. For example, in a melanoma model, EGCG was found to drive expression of let7-b, an miRNA that is important for inducing apoptosis through targeting HMGA2, a driver of tumor progression [102]. The mechanism by which EGCG induces miRNA expression is unknown, as it may impact different miRNAs through different mechanisms. Generally, EGCG could be functioning to drive transcription, or to facilitate miRNA processing events required for their function. MiRNA-210 provides an example of a known regulatory mechanism facilitated by EGCG. In both human and mouse models of lung cancer, it was found that EGCG drove expression of miRNA-210 [103]. MiRNA-210 contains a hypoxia response element promoter and is thus regulated by HIF-1a in hypoxic conditions [103]. EGCG was shown to directly interact with the oxygen-dependent degradation (ODD) domain of hypoxia inducible factor-1a (HIF-1a) in order to stabilize it to drive miRNA-210 transcription [103]. It is important to note that miRNA expression is not consistent across cancer types. For example, in a melanoma model, EGCG did not impact miRNA-210 expression, suggesting this pathway may not be relevant in all cancer types [102].

## 5. Role of Green Tea against Reproductive Cancers

### 5.1. EGCG and Ovarian Cancer

Ovarian cancer is the second most common gynecologic cancer in the US and is the fifth leading cause of cancer-related deaths, with epithelial ovarian cancer being the most common subtype [104,105]. Although advances in treatment options have been made throughout the years, ovarian cancer is still associated with a high burden of disease and poor prognosis [106]. Risk factors include nulliparity, early menarche, obesity, and family history of breast or ovarian cancer [107]. Ovarian cancer tumorigenesis is due to aberrations in the PI3K/AKT/mTOR signaling pathway [108]. Studies have demonstrated that mutations or amplifications in PIKC3CA, mutations in PIK3R, loss of PTEN, and mutations or amplifications in AKT isoforms cause uncontrolled activation of this pathway and ovarian cancer formation [109]. Green tea, with its main bioactive component, EGCG, has anti-cancer properties, and emerging evidence from in vitro and in vivo studies suggests a decreased risk of ovarian cancer progression with green tea consumption [106]. EGCG has been characterized with antiproliferative, antiangiogenic, and antimetastatic properties [110]. It also promotes apoptosis and autophagy of ovarian cancer cells [110].

In vitro and in vivo studies have focused on the mechanisms behind the anti-cancer properties associated with green tea extracts, especially EGCG. It has been shown that the latter inhibits cell proliferation, infiltration, and metastasis, and induces apoptosis of ovarian cancer cells in a time- and dose-dependent manner [111]. Cell cycle proliferation is inhibited by targeting the PTEN/AKT/mTOR pathway in vivo and in vitro [112]. Treatment of SKOV3 cells with EGCG (20, 40 µg/mL) resulted in the upregulation of PTEN, which leads to dephosphorylation of phosphatidylinositol-3,4,5-triphosphate (PIP3) to phosphatidylinositol 4,5-bisphosphate (PIP2) and subsequently to the inhibition of AKT and mTOR [112]. The inhibition of the AKT/mTOR pathway was shown to induce apoptosis of cancer cells [112]. Additionally, EGCG’s proapoptotic properties are attributed to the increased expression of pro-apoptotic proteins p21, Bax, and caspase-3, and to the decreased expression of anti-apoptotic proteins PCNA, Bcl-2, and Bcl-Xl [112,113]. Treatment with EGCG (20–100 µg/mL) also inhibited cancer cell infiltration and metastasis by downregulating aquaporin 5 (AQP5), NF-κB, and p65 [114]. Treatment of HEY and OVCA433 cells with 20 to 40 µmol/L of EGCG treatment resulted in the downregulation of ET_A_R-dependent signaling pathways, including MAPK and PI3K/AKT pathways. ET_A_R inhibition also resulted in the downregulation of matrix metalloproteinases expression, which prevents cancer cell invasion and metastasis [115] (Table 1) (Figure 5).

Ovarian cancer regression was also observed in female athymic mice treated with EGCG (12.4 g/L) [115]. Hence, EGCG exerts its anti-cancer properties by targeting a range of signaling pathways, particularly the PI3K/AKT/mTOR pathway, which is responsible for uncontrolled cancer cell proliferation and metastasis (Table 2). Further studies have also shown that dysregulated autophagy plays a role in ovarian cancer pathogenesis. Disruption in autophagy regulators such as LC-3BII, p62, ATG5, Beclin1, and ULK1 induced ovarian cancer by promoting cellular growth and preventing autophagosomes formation [116].

Green tea has also been used in conjunction with several chemotherapeutic agents. This combination has been shown to have a synergistic effect on the treatment of ovarian cancer cells. For instance, a synergistic effect was observed in the induction of apoptosis in ovarian cancer cell lines, SKOV-3 and OVCAR-3, when green tea extract (25, 50 µg/mL) was combined with Paclitaxel (20, 40 µg/mL), a chemotherapy drug that blocks microtubules’ function [117]. Combined treatment blocked the phosphorylation of AKT, enhanced the expression of pro-apoptotic proteins (mainly Bax, CytC, and caspase-3 and 9), and decreased the expression of anti-apoptotic proteins such as Bcl-2 [117]. It was found that combining EGCG (10 µM) with Cisplatin would enhance the sensitivity of ovarian cancer cells to the chemotherapeutic agent by inducing copper transporter 1 (CTR1) protein expression and by blocking its degradation in OVCAR3 and SKOV3 cells as well as xenograft mice. This transporter is responsible for the intracellular uptake of Cisplatin [118].

Several epidemiological studies examined the effect of green tea consumption on epithelial ovarian cancer occurrence. Among them, two case-control studies showed that daily high-dose consumption of green tea was inversely associated with epithelial ovarian cancer occurrence (OR: 0.43, 95% CI: [0.30; 0.63]) [119], (OR: 0.46, 95% CI [0.26; 0.84]) [120]. However, two other studies did not find any association between the two (OR: 0.90, 95% CI [0.50; 1.61]) [121], (OR: 0.82, 95% CI [0.38; 1.79]) [122], but a meta-analysis did confirm the presence of a statistically significant inverse relationship (OR: 0.66, 95% CI [0.54; 0.80]) [123]. Yet, another study assessed the progression of disease and survival rate in women who consumed green tea and diagnosed with epithelial ovarian cancer. Women who drank at least one cup of green tea per day in that study had a higher survival rate compared to non-drinkers (HR: 0.55 (95% CI = 0.34–0.90)) and a direct dose response relationship was significant with higher survival rates associated with increased dose of green tea consumption [124] (Table 3). A phase II clinical trial was conducted to study the effect of green tea on ovarian cancer recurrence. Catechin enriched green tea was used (500 mL daily) and its use in that clinical scenario was not promising despite high adherence rate to the drug intake: only 5 out of 16 women remained disease-free after 18 months (NCT00721890) (Table 4).
cancers-15-00862-t001_Table 1Table 1In vitro studies examining the effect of green tea on reproductive cancers.
Study and YearCell LinesGreen Tea Extract/EGCG ConcentrationsMolecular TargetsMechanisms of ActionOvarian CancerRao et al. (2010) [111]SKOV-310–80 μg/mLDNA fragmentation Cell morphological changesCell cycle arrestInhibition of cellular proliferationInduction of apoptosisQin et al. (2020) [112]SKOV-3, CAOV-3, NIH-OVCAR-30, 5, 10, 20, 40, and 80 μg/mLInhibiton of PTEN/AKT/Mtor signaling pathway Upregulation of caspase-3 and Bax Downregulation of Bcl-2Inhibition of cellular proliferationInduction of apoptosisSeung et al. (2004) [113]SKOV-3, OVCAR-3, PA-16.25, 12.5, 25, 50, 100 μMCell cycle arrest in G1 phaseIncreased expression of p21WAF, BaxDecreased expression of PCNA, and Bcl-XlInhibiton of cellular proliferationInduction of apoptosisYan et al. (2012) [114]SOV-320–100 μg/mLDownregulation of p65, AQP5, NF-ΚbInhibition of cellular proliferationInduction of apoptosisSpinella et al. (2006) [115]HEY, OVCA-43320–40 μMDecreased expression of ET_A_R and ET-1Reduction in ET_A_R-dependent signaling pathwaysInhibition of cellular proliferationInduction of apoptosisCervical CancerZou et al. (2010) [125]HeLa, Me180, TCL11, 5, 10, 25, 50 μg/mLIncreased expression of p53 and p21Inhibition of cellular proliferationInduction of apoptosisSingh et al. (2011) [126]HeLa10, 15, 20 μg/mLCell cycle arrest in G1 phaseIncrease in reactive oxygen species, p53 and Bax expression, cytochrome c releaseInhibition of AKT, NF-Κb and cyclin D1Inhibition of cellular proliferationInduction of apoptosisKuhn et al. (2003) [127]HeLa10 μMInhibition of ubiquitin/proteasome protein degradationIncrease in p27 and Bax proteinsInduction of apoptosisBonfili et al. (2011) [128]HeLa0–140 μMInhibition of proteasomeAccumulation of p53, p27 and IκB-αInduction of apoptosisLi et al. (2007) [129]HeLa2.5–20 μMInhibition of IGF-IRInhibition of cellular proliferation and anchorage independent cell transformationChakrabarty et al. (2015) [130]HeLa10–50 μMDepolymerization of cellular microtubulesInhibition of cellular proliferationAhn et al. (2003) [131]CaSki35 μMCell cycle arrest at G1Inhibition of cellular proliferationInduction of apoptosisQiao et al. (2009) [132]CaSki, HeLa25, 50 μMInhibition of HPV E6 and E7Decrease ER-α and aromatase expressionInhibition of cellular proliferationSah et al. (2004) [133]HeLa, CaSki, SiHa5–50 μMInhibition of EGFR, ERK1/2 and AKT activationIncrease in p53, p21, p27Decrease in CDK2 kinase activityInhibition of cellular proliferationInduction of apoptosisZhang et al. (2006) [134]HeLa10–100 μmol/LInhibition of HIF-1 α protein accumulationVEGF expressionInhibition of PI3K/AKT and ERK1/2 signaling pathwaysInhibition of angiogenesisTudoran et al. (2012) [135]HeLa10 μMDownregulation of PDFA and TGF-β_2_Upregulation of IL-1 βMaintenance of cellular morphologyInhibition of cellular proliferationInhibition of angiogenesisInhibition of metastasisSharma et al. (2012) [136]HeLa25 μMDecreased expression of MMP-9Increased expression of TIMP-1Inhibition of cellular proliferationInduction of apoptosisInhibition of metastasisRoomi et al. (2010) [137]HeLa, DoTc2-451010–100 μMInhibition of MMP-2 and MMP-9 expressionInhibition of metastasisPanji et al. (2021) [138]HeLa, SiHa0–100 μmol/LInhibition of TGF- β induced epithelial to mesenchymal transitionInhibition of metastasisEndometrial CancerManohar et al. (2013) [139]Ishikawa100–150 μMDecreased activation of ERKUpregulation of Bax and downregulation of bcl-2Increase ROS and p38 activationInhibition of cellular proliferationInduction of apoptosis
Park et al. (2012) [140]Ishikawa50, 100 μMCell cycle arrestInhibition of MAPK and AKT signaling pathwaysIncrease Bax/bcl ratioInhibition of cellular proliferationInduction of apoptosis
Man et al. (2020) [141]RL95-2, AN3 CA20, 40, 60 μM (Pro-EGCG)Activation of p38 MAPKInhibition of AKT/ERK signaling pathwayInhibition of cellular proliferationInduction of apoptosis
Wang et al. (2018) [142]RL95-2, AN3 CA20, 40, 60 μM (Pro-EGCG)Decrease VEGF by inhibiting PI3K/AKT/mTOR/HIF-1 α signaling pathwayInhibition of angiogenesisVulvar CancerYap et al. (2021) [143]HFK-HPV18, VIN cl.11100 μMDownregulation of E6 and E7 expressionInhibition of cellular proliferation
cancers-15-00862-t002_Table 2Table 2In vivo studies examining the effect of green tea on reproductive cancers.
Study and YearAnimal ModelGreen Tea Extract/EGCG ConcentrationsMolecular TargetsMechanisms of ActionOvarian CancerQin et al. (2020) [112]Female BALB/c nude mice10, 30 or 50 mg/kgInhibition of PTEN/AKT/mTOR signaling pathwayInhibition of ovarian tumor growthSpinella et al. (2006) [115]Female athymic (nu^+^/nu^+^) mice12.4 g/LDecreased expression of ET_A_R and ET-1Inhibition of ovarian tumor growthCervical CancerRoomi et al. (2015) [144]Female nude mice0.5% supplementation with dietary mixtureIncrease in extracellular matrix proteinsInhibition of tumor growthRoomi et al. (2015) [145]Female athymic nude mice0.5% supplementation with dietary mixtureDecreased MMP-2 and MMP-9, Bcl-2 and VEGF expressionInhibition of proliferationInhibition of angiogenesisInhibition of metastasisEndometrial CancerMan et al. (2020) [141]Female athymic nude mice50 mg/kgInhibition of anti-apoptotic molecules NOD1 and NAIPInhibition of tumor growth
cancers-15-00862-t003_Table 3Table 3Observational studies examining the effect of dietary intake of green tea on reproductive cancers.
Study and YearType of StudyCases/SizeExposure vs. ControlOR (95% CI)Ovarian CancerNagle et al. (2010)[122]Case-control1271/1198Never vs. ≥1 cup/day0.82 (0.38–1.79)Zhang et al. (2002)[119]Case-control254/652Never or rarely vs. ≥1 time/day0.43 (0.30–0.63)Song et al. (2008)[120]Case-control781/1263Never or rarely vs. ≥1 cup/day0.46 (0.26–0.84)Goodman et al. (2003)[121]Case-control164/194Never vs. ≥1 cup/day0.90 (0.50–1.61)Endometrial CancerGao et al. (2005)[146]Case-control995/1087Never vs. ≥7 cups/day0.76 (0.60–0.95)Kakuta et al. (2009)[147]Case-control152/2854 cups/week vs. ≥4 cups/day0.33 (0.15–0.75)Hirose et al. (2007)[148]Case-control229/2425Never vs. ≥7 cups/day1.33 (0.75–2.35)Shimazu et al. (2008)[149]Prospective cohort117/53,7244 cups/week vs. ≥5 cups/day0.75 (0.44–1.30)Xu et al. (2007)[150]Case-control1204/1212Never vs. ever green tea0.80 (0.60–0.90)
cancers-15-00862-t004_Table 4Table 4Interventional trials studying the effect of green tea on reproductive cancers.
StudyPhaseIntervention vs. ControlLength of InterventionResultsOvarian Cancer(FIGO stage III-IV serous or endometrioid ovarian cancer)Trudel et al. [151]NCT00721890Phase II clinical trialN = 16Nonrandomized, single-arm, two-stage design. All treated with double-brewed green tea 500 mL (~639 mg/mL EGCG) for up to 18 monthsPrimary outcome absence of recurrence at 18 monthsGreen tea intake for minimum less than 100 days; maximum more than 3 yearsAbsence of recurrence of ovarian cancer 5/16 women at 18 monthsTrial terminatedCervical Intraepithelial Neoplasia (CIN)Garcia et al. [152]NCT00303823Phase II clinical trialN = 9850 treated and 48 control(41 in each group were analysed)Randomized to Polyphenon E (800 mg of EGCG) vs. placeboPrimary outcomes: HPV clearance or CIN1 resolution at 4 monthsPolyphenon E or placebo intake for 16 weeksFollow up after 2 weeks of treatment completionClearance was similar. Progression of cervical lesion in 14.6% of women receiving intervention vs. 7.7% in those receiving placeboAhn et al.[153]Clinical trialN = 5127 treated and 39 controlRandomized to Polyphenon E and EGCG ointment or 200 mg capsule vs. untreated control groupPrimary outcomes: HPV DNA titers, histology or cytologyPolyphenol E ointment or 200 mg capsule or 200 mg EGCG capsule taken for 8 to 12 weeks69% (35/51 patients)response in group taking green tea extract vs. 10% (4/39 patients) response in control groupVulvar Usual type of vulvar Intraepithelial neoplasia (uVIN)Yap et al.[154]Phase II clinical trialN = 2613 treated and 13 controlRandomized 1:1 to Sinecatechins 10% vs. placebo ointmentPrimary outcome was histological resolution of uVIN at 32 weeksSinecatechins ointment or placebo for 16 weeksFollow up at 2, 4, 8, 16, 32 and 52 weeks5/13 patients who received sinecatechins had complete clinical response and 8/13 had partial response


### 5.2. EGCG and Cervical Cancer

Cervical cancer is both the fourth most common and leading cause of cancer-related deaths among women [155]. Oncogenic strains of HPV16 and HPV18 infections are proven causal agents. Other risk factors include smoking, history of multiple sexual partners, unprotected intercourse, and sexual activity at early age [156]. Although prevention of this cancer is available with HPV vaccination, advances in active disease treatment are still needed. Green tea with its major active moiety EGCG and attributed anti-cancer properties has been proven to be beneficial in the treatment of cervical cancer.

In vitro studies have examined the effects of green tea on cervical cancer treatment. EGCG inhibited cellular proliferation by inducing cell cycle arrest in a dose- and time- dependent manner in different cervical cancer cell lines, including Me180, HeLA, CaSki, and SiHa cells [125,126,131]. EGCG also suppressed HPV oncogene and oncoprotein expression, particularly for E6 and E7. E6 oncoprotein is known to degrade the tumor suppressor gene p53 via the ubiquitin-proteasome pathway, and E7 oncoprotein is known to induce retinoblastoma tumor suppressor gene expression pRb degradation [157]. EGCG stopped the proliferation of cervical cancer cells and induced their apoptosis by blocking the activity of E7 protein. The mechanism behind E7 blockage lies on restraining the ubiquitin-proteasomal activity and inhibiting tumor growth by accumulating proteasomes targets and ubiquitinated proteins, including p53, p27, and IκB-α in HeLa cells [127,128,158]. EGCG (25, 50 µM) also inhibited aromatase and estrogen receptor α (ERα) in cervical cancer cells, which would limit the expression of E6 and E7 [132]. Inhibition of E6 by EGCG would result in the upregulation of p53, which would block growth factor receptor signaling pathways, especially EGFR and ERK1/2/AKT. This contributes to further inhibition of cellular proliferation [133]. Furthermore, EGCG (5–20 µM) also reduced activity of IGF-1R and reduced its affinity to its ligand insulin, such as growth factor 1 (IGF-1), thus blocking anchorage-independent transformation of HeLa cells [129]. EGCG’s proapoptotic properties lie in the blockage of tubulin assembly and in the induction of microtubule depolymerization by binding to the tubulin α subunit at a concentration of 50 µM [130]. Additionally, it induced the expression of pro-apoptotic molecules such as Bax and caspase-3 [126,127].

EGCG is not only characterized by anti-proliferative and proapoptotic properties, but also has anti-angiogenic and anti-metastatic ones. In HeLa cells, the green tea compound (100 µmol/L) suppressed the accumulation of hypoxia-induced HIF-1α protein by blocking AKT and ERKs signaling pathways and allowed for the degradation of HIF-1α by the proteasome. Therefore, VEGF levels, which are dependent on HIF-1α drop, are decreased [134]. Blocking AKT/ERK signaling pathways lead to the downregulation of matrix metalloproteinases, especially MMP-2 and MMP-9, and to the upregulation of the tissue inhibitors of metalloproteinases 1 (TIMP-1) expression in HeLa cells [136,137]. EGCG also exerted its anti-metastatic properties by inhibiting the epithelial-to-mesenchymal transition mediated by TGF-β via ROS/smad signaling pathway in HeLa and SiHa cells at a concentration of 60 (µmol/L) [138] (Table 1) (Figure 6).

In vivo studies also investigated the effect of EGCG on cervical cancer treatment. For instance, Roomi et al. found that administration of nutrient mixture containing green tea extract for 4 weeks to female athymic nude mice led to a significant decrease in cervical cancer growth (Table 2) [144,145].

A combination of green tea extract and chemotherapeutic agents has been used in the treatment of cervical cancer. For instance, a synergistic effect in inhibition HeLa cells proliferation was observed while combining Cisplatin (250 nM) with EGCG (25 µM) [159]. Additionally, retinoic acid (1 µM) in conjunction with EGCG (50–100 µM) suppressed telomerase activity and induced apoptosis [160]. The combination of tea polyphenols (25–125 µg/mL) and bleomycin (5–25 µM/mL), a chemotherapeutic agent responsible for DNA oxidative damage, also showed a synergistic increase in caspase-3,-8,-9 activation as well as p53 activation [161]. Moreover, the combination of EGCG and doxorubicin promoted autophagic cell death by increasing activity of autophagy modulators [162].

Several randomized controlled trials yielded conflicting results for the use of green tea extract in preventing or treating cervical lesions. A randomized phase II trial studying the effect of green tea extract, Polyphenon E, in patients with HPV infection and low-grade cervical intraepithelial neoplasia (CIN) showed no effect on cervical cancer prevention because there was no infection clearance or lesion regression (NCT00303823). However, a trial evaluating the role of Polyphenol E and EGCG in the form of ointment or 200 mg capsule in women with HPV infected cervical lesions did show an effective response in treating such lesions [153] (Table 4).

### 5.3. EGCG and Endometrial Cancer

Endometrial cancer is the third most common cause of death-related cancer in women. The major risk factor is unopposed estrogen exposure. Other risks include early menarche and tamoxifen exposure [163]. Cancer development is due to uncontrolled proliferation mediated by unopposed estrogen exposure and by the mismatch repair (MMR) system. Specific to type 1 endometrial cancer are mutations in PTEN, K-ras, and β-catenin, while specific to type 2 endometrial cancer are mutations in p53 and HER-2/neu [164]. Several studies examined the role of green tea in vitro and in vivo for treating endometrial cancer.

EGCG’s antiproliferative effect on endometrial cancer cell HEK-293 and Ishikawa cells was due to the inhibition of the PI3K/AKT and MAPK signaling pathways. A proapoptotic effect was observed by the increase in pro-apoptotic protein expression including caspase-6,8,10 and Bax, and by the decrease in anti-apoptotic protein expression Bcl-2 and Bcl-XL [139,140]. Autophagy of endometrial cancer cells was also activated by blocking the PI3K/AKT/mTOR pathway [165].

Pro-EGCG, which is a prodrug of EGCG, is characterized by enhanced stability and improved bioavailability. The prodrug also exhibited anti-tumor properties in a time- and dose-dependent manner (20, 40, 60 µM). Its anti-proliferative effect was attributed to the increased phosphorylation of p38 MAPK and JNK and to the decreased phosphorylation of AKT and ERK [141]. Pro-EGCG also inhibited tumor growth in vivo by decreasing anti-apoptotic molecules NOD1 and NAIP [141]. The anti-angiogenic effects were related to the downregulation of VEGF and HIF-1a in RL95-2, PHES and AN3 CA EC cells [166] (Table 1 and Table 2).

Several epidemiological studies examined the effect of green tea in treating endometrial cancer. A case-control study found an inverse relationship between drinking green tea and endometrial cancer occurrence in premenopausal women in a dose-dependent manner (OR: 0.74, 95% CI: 0.54–1.01) [146]. Two meta-analyses found similar trends in results. The first one reported an 11% decrease in risk of cancer incidence [167], while the second found a risk ratio of 0.79, with a 95% CI 0.69–0.90 [168]. Several studies also found a similar dose response relationship between green tea consumption and the risk of endometrial cancer, but other studies did not [146,147,148,149,150,169] (Table 3).

### 5.4. EGCG and Vulvar Cancer

Vulvar cancer represents approximately 6% of all gynecologic cancers [170]. The major risk factor for vaginal and vulvar cancer is infection with oncogenic HPV strains, mostly HPV 16 and HPV 18.

A study explored the role of EGCG treatment on the HPV18-positive vulvar intraepithelial neoplasia (VIN) clone on cell proliferation and viral replication [143]. The investigators observed that EGCG (100 µM) inhibited cellular proliferation by decreasing viral oncoproteins E6 and E7. EGCG did so not by altering their expression but rather by increasing cellular protein turnover through the proteasome in both HFK-HPV18 and VIN cl.11 cells. Decreased levels of E6 and E7 resulted in an increase in their associated tumor suppressor genes and arrest of cellular proliferation. However, EGCG treatment did not affect keratinocyte differentiation (Table 1).

A phase II clinical trial researching the effect of Sinecathetin ointment 10%, which contains EGCG, in women with vulvar intraepithelial neoplasia showed improvement of symptoms and lesion regression [154] (Table 4).

## 6. Conclusions and Future Perspectives

In summary, green tea, with its main bioactive compound EGCG, has proven benefits in the treatment of reproductive cancers, including ovarian, cervical, endometrial, vaginal, and vulvar cancers. In vitro studies indicated that EGCG may act on several receptors and intracellular signaling pathways involved in cancer formation and survival. In vivo studies further confirmed the benefits of EGCG in tumor prevention and shrinkage, though the results of clinical trials have been mixed. Although treatments involving EGCG remain controversial at this time, there is evidence suggesting a beneficial effect of green tea in the treatment of reproductive cancers. However, more studies are needed to draw definitive conclusions and stronger recommendations for potential use of green tea, alone or in combination with chemotherapeutic agents, in cancer therapy. Notably, the role of many EGCG-interacting proteins discussed in this review has not been fully characterized in gynecological cancers. Given EGCG’s effects as both a pro- and antioxidant, there are unique opportunities for its use in the treatment of cancer that are dependent on the progression of the disease. This review highlights many of the mechanisms by which EGCG may be affecting the body, and thus provides direction for further research. Because of its wide accessibility and abundance as it is a natural compound, it is important to continue investigating EGCG as a promising therapeutic.

## Figures and Tables

**Figure 1 cancers-15-00862-f001:**
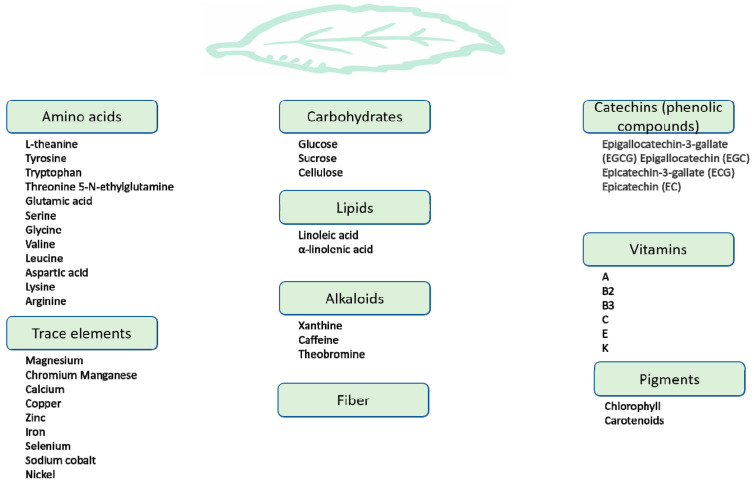
The chemical composition of green tea.

**Figure 2 cancers-15-00862-f002:**
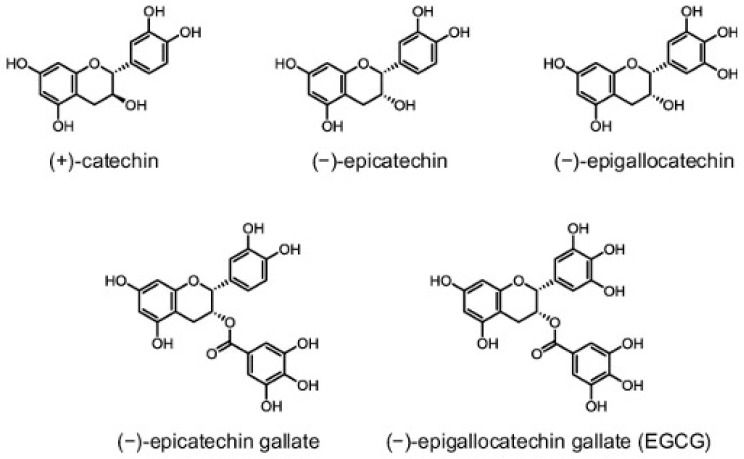
Chemical structure of catechins.

**Figure 3 cancers-15-00862-f003:**
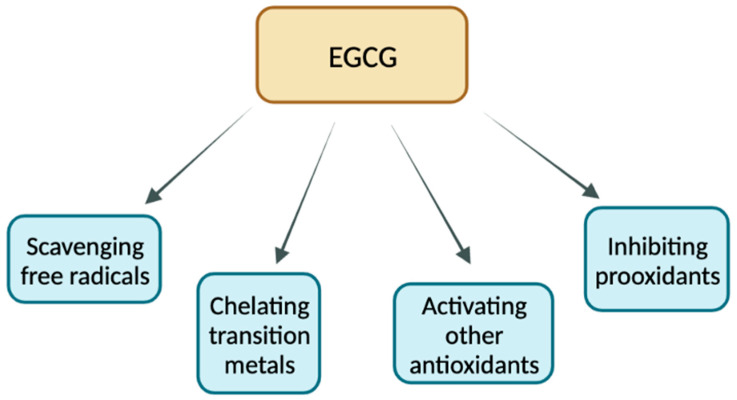
Antioxidant effects of EGCG.

**Figure 4 cancers-15-00862-f004:**
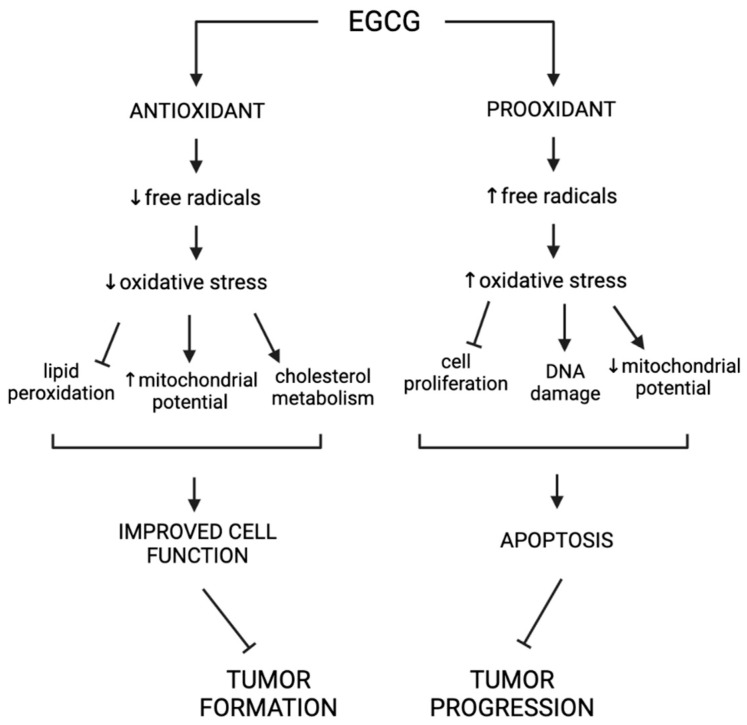
Pro- and antioxidant functions of EGCG.

**Figure 5 cancers-15-00862-f005:**
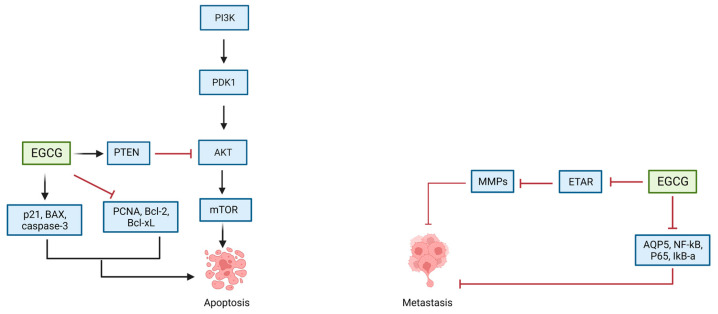
Effect of EGCG on ovarian cancer cells.

**Figure 6 cancers-15-00862-f006:**
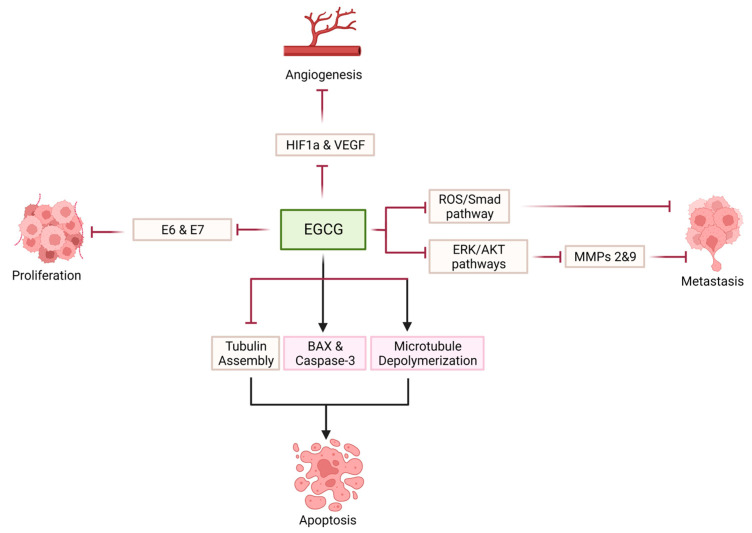
Effect of EGCG on cervical cancer cells.

## Data Availability

The data presented in this study are available in the article and tables.

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
