# Peer review of "Green Tea in Reproductive Cancers: Could Treatment Be as Simple?"

_cancers, 2023, doi:10.3390/cancers15030862_

Round 1

Reviewer 1 Report

Dear editor, thank you for giving me this opportunity to review this study.

I read with great interest, it is valuable study, which tries to summarize previously result regarding the green tea and its impact on prevention of reproductive cancer. Here are my comments:

Is this study a narrative review? If yes, please mention it in title and abstract of study.

Is figure 1 original? If no, please cite to the reference.

Figure 1. Please see the figure, alkaloids is made of three items xanthine, caffeine, and theobromine, but in the legend of figure it is written (caffeine, theophylline and theobromine), please use the same word or in the figure instead of xanthine write 1,3-dimethylxanthine.

There is no need all item at the figure are repeated at the legend. Please re-summarize the legend of figure.

MMP- 2, and MMP-9, HSP90, PIP, E6,…..

 … Please before using abbreviation introduce the full text of them.

Role of Green Tea Against Reproductive Cancers:

Regarding the epidemiology of the cancers, please use the most updated references. In ovarian cancer 2017????

Please draw (a) table(s) and summarize the characteristics/result of previously published studies regarding the impact of green tea on reproductive cancers or with its mechanism of action on the reproductive cancers.

“Among them, three case-control studies showed that 503 daily high dose consumption of green tea was inversely associated with epithelial ovarian 504 cancer occurrence (OR: 0.82, 95% CI [0.38; 1.79]) [105], (OR: 0.43, 95% CI: [0.30; 0.63]) [106], 505 (OR: 0.46, 95% CI [0.26; 0.84]) [107]”

Based on these sentence just in two study there was a relationship. Study with (OR: 0.82, 95% CI [0.38; 1.79]): relationship in cite 105 was not significant, because 95% CI include the number 1. Please revise the above mentioned sentences.

Please revise the Table 1. Add the references of each study in the table 1.  2. Omit the PMID studies, 3. Include the other characteristics of studies like year, sample size, type of studies, ..

Author Response

Response: Manuscript ID cancers-2165764

Dear Editor,

We thank you and the reviewers for their helpful suggestions regarding the manuscript entitled: “Green Tea in Reproductive Cancers: Could Treatment be as Simple?”

We have addressed each of the reviewers’ comments point-by-point in the comments below. In the revised manuscript, altered text is shown by “track changes”. The authors believe the suggestions of the reviewers have greatly improved the manuscript.

We hope these changes will be satisfactory and look forward to any further suggestions.

Best wishes,

Md Soriful Islam,

for the authors

Response to Reviewer 1 comments:

Reviewer 1

Comments and Suggestions for Authors

Dear editor, thank you for giving me this opportunity to review this study.

I read with great interest, it is valuable study, which tries to summarize previously result regarding the green tea and its impact on prevention of reproductive cancer. Here are my comments:

Is this study a narrative review? If yes, please mention it in title and abstract of study.

Author’s response: Thank you for your concern. This is a “scoping review”. We appreciate your suggestion and we revised the texts in the abstract accordingly.

Changes in abstract are: A comprehensive search of PubMed and Google Scholar up to December 2022 was conducted. All original and review articles related to green tea or EGCG, and gynecological cancers published in English were included.

Is figure 1 original? If no, please cite to the reference.

Author’s response: Thank you for this question. Figure 1 is original.

Figure 1. Please see the figure, alkaloids is made of three items xanthine, caffeine, and theobromine, but in the legend of figure it is written (caffeine, theophylline and theobromine), please use the same word or in the figure instead of xanthine write 1,3-dimethylxanthine.

Author’s response: Thank you for pointing that out. We changed the wording in the text to match the figure labels as suggested.

Changes in manuscript (figure legends) are: Caffeine, xanthine and theobromine.

There is no need all item at the figure are repeated at the legend. Please re-summarize the legend of figure.

Author’s response: Thank you for the suggestion. The figure legends have been re-summarized to avoid redundancy.

Changes in manuscript (figure legends) are:

Figure 3. Antioxidant effects of EGCG

Figure 4. Pro and antioxidant functions of EGCG

Figure 5. Effect of EGCG on ovarian cancer cells

Figure 6. Effect of EGCG on cervical cancer cells

MMP- 2, and MMP-9, HSP90, PIP, E6,…..

 … Please before using abbreviation introduce the full text of them.

Author’s response: Thank you for recognizing this inconsistency. Abbreviations throughout the manuscript have been identified by full name before using them in the updated text.

Role of Green Tea Against Reproductive Cancers:

Regarding the epidemiology of the cancers, please use the most updated references. In ovarian cancer 2017????

Author’s response: Thank you for pointing out this discrepancy. The most updated data regarding the epidemiology of the reproductive cancers is now used.

Changes in manuscript are: Ovarian cancer is the second most common gynecologic cancer in the US and is the fifth leading cause of cancer related deaths with epithelial ovarian cancer being the most common subtype [104,105].

Vulvar cancer represents approximately 6% of all gynecologic cancers [170].

Please draw (a) table(s) and summarize the characteristics/result of previously published studies regarding the impact of green tea on reproductive cancers or with its mechanism of action on the reproductive cancers.

Author’s response: Thank you for this great suggestion. Two tables (tables 1 and 2) have been added about the mechanism of action of EGCG on reproductive cancers in vitro and in vitro.

Changes in manuscript are:

Study & Year

Cell Lines

Green tea extract/ EGCG Concentrations

Molecular  targets

Mechanisms of Action

Ovarian  Cancer

Rao et al. (2010) [112]

SKOV-3

10-80 μg/mL

DNA fragmentation Cell morphological changes

Cell cycle arrest

Inhibition of cellular proliferation

Induction of apoptosis

Qin et al. (2020) [113]

SKOV-3, CAOV-3, NIH-OVCAR-3

0, 5, 10, 20, 40, and 80 μg/mL

Inhibiton of PTEN/AKT/mTOR signaling pathway Upregulation of caspase-3 and Bax Downregulation of Bcl-2

Inhibition of cellular proliferation

Induction of apoptosis

Seung et al. (2004) [114]

SKOV-3, OVCAR-3, PA-1

6.25, 12.5, 25, 50, 100 μM

Cell cycle arrest in G1 phase

Increased expression of p21WAF, Bax, Decreased expression of PCNA, and Bcl-Xl

Inhibiton of cellular proliferation

Induction of apoptosis

Yan et al. (2012) [115]

SOV-3

20-100 μg/mL

Downregulation of p65, AQP5, NF-Κb

Inhibition of cellular proliferation

Induction of apoptosis

Spinella et al. (2006) [116]

HEY, OVCA-433

20-40 μM

Decreased expression of ETAR and ET-1

Reduction of ETAR-dependent signaling pathways

Inhibition of cellular proliferation

Induction of apoptosis

Cervical  Cancer

Zou et al. (2010) [126]

HeLa, Me180, TCL1

1, 5, 10, 25, 50 μg/mL

Increased expression of p53 and p21

Inhibition of cellular proliferation

Induction of apoptosis

Singh et al (2011) [127]

HeLa

10, 15, 20 μg/mL

Cell cycle arrest in G1 phase

Increase in reactive oxygen species, p53 and Bax expression, cytochrome c release

Inhibition of AKT, NF-Κb and cyclin D1

Inhibition of cellular proliferation

Induction of apoptosis

Kuhn et al. (2003) [128]

HeLa

10 μM

Inhibition of ubiquitin/proteasome protein degradation

Increase in p27 and Bax proteins

Induction of apoptosis

Bonfili et al. (2011) [129]

HeLa

0-140 μM

Inhibition of proteasome and accumulation of p53, p27 and IκB-α

Induction of apoptosis

Li et al. (2007) [130]

HeLa

2.5-20 μM

Inhibition of IGF-IR

Inhibition of cellular proliferation and anchorage independent cell transformation

Chakrabarty et al. (2015) [131]

HeLa

10-50 μM

Depolymerization of cellular microtubules

Inhibition of cellular proliferation

Ahn et al. (2003) [132]

CaSki

35 μM

Cell cycle arrest at G1

Inhibition of cellular proliferation

Induction of apoptosis

Qiao et al. (2009) [133]

CaSki, HeLa

25, 50 μM

Inhibition of HPV E6 and E7

Decrease ER- α and aromatase expression

Inhibition of cellular proliferation

Sah et al. (2004) [134]

HeLa, CaSki, SiHa

5-50 μM

Inhibition of EGFR, ERK1/2 and AKT activation

Increase in p53, p21, p27

Decrease in CDK2 kinase activity

Inhibition of cellular proliferation

Induction of apoptosis

Zhang et al. (2006) [135]

HeLa

10-100 μmol/L

Inhibition of HIF-1 α protein accumulation

VEGF expression, inhibition of PI3K/AKT and ERK1/2 signaling pathways

Inhibition of angiogenesis

Tudoran et al. (2012) [136]

HeLa

10 μM

Downregulation of PDFA and TGF-β2

Upregulation of IL-1 β

Maintenance of cellular morphology

Inhibition of cellular proliferation

Inhibition of angiogenesis

Inhibition of metastasis

Sharma et al. (2012) [137]

HeLa

25 μM

Decreased expression of MMP-9 Increased expression of TIMP-1

Inhibition of cellular proliferation

Induction of apoptosis   Inhibition of metastasis

Roomi et al. (2010) [138]

HeLa, DoTc2-4510

10-100 μM

Inhibition of MMP-2 and MMP-9 expression

Inhibition of metastasis

Panji et al. (2021) [139]

HeLa, SiHa

0-100 μmol/L

Inhibition of TGF- β induced epithelial to mesenchymal transition

Inhibition of metastasis

Endometrial Cancer

Manohar et al. (2013) [140]

Ishikawa

100-15 μM

Decreased activation of ERK

Upregulation of Bax and downregulation of bcl-2

Increase ROS and p38 activation

Inhibition of cellular proliferation

Induction of apoptosis

Park et al. (2012) [141]

Ishikawa

50, 100 μM

Cell cycle arrest

Inhibition of MAPK and AKT signaling pathways

Increase BAX/bcl ratio

Inhibition of cellular proliferation

Induction of apoptosis

Man et al. (2020) [142]

RL95-2, AN3 CA

20, 40, 60 μM (Pro-EGCG)

Activation of p38 MAPK

Inhibition of AKT/ERK signaling pathway

Inhibition of cellular proliferation

Induction of apoptosis

Wang et al. (2018) [143]

RL95-2, AN3 CA

20, 40, 60 μM (Pro-EGCG)

Decrease VEGF by inhibiting PI3K/ AKT/ mTOR/ HIF-1 α signaling pathway

Inhibition of angiogenesis

Vulvar   Cancer

Yap et al. (2021) [144]

HFK-HPV18, VIN cl.11

100 μM

Downregulation of E6 and E7 expression

Inhibition of cellular proliferation

Table 1. In vitro studies examining the effect of green tea on reproductive cancers

Study & Year

Animal Model

Green tea extract/ EGCG Concentrations

Molecular Targets

Mechanisms of Action

Ovarian Cancer

Qin et al. (2020) [113]

Female BALB/c nude mice

10, 30 or 50 mg/kg

Inhibition of PTEN/AKT/mTOR signaling pathway

Inhibition of ovarian tumor growth

Spinella et al. (2006) [116]

Female athymic (nu+/nu+) mice

12.4 g/L

Decreased expression of ETAR and ET-1

Inhibition of ovarian tumor growth

Cervical Cancer

Roomi et al. (2015) [145]

Female nude mice

0.5% supplementation with dietary mixture

Increase in extracellular matrix proteins

Inhibition of tumor growth

Roomi et al. (2015) [146]

Female athymic nude mice

0.5% supplementation with dietary mixture

Decreased MMP-2 and MMP-9, Bcl-2 and VEGF expression

Inhibition of proliferation

Inhibition of angiogenesis

Inhibition of metastasis

Endometrial Cancer

Man et al. (2020) [142]

Female athymic nude mice

50 mg/kg

Inhibition of anti-apoptotic molecules NOD1 and NAIP

Inhibition of tumor growth

Table 2. In vivo studies examining the effect of green tea on reproductive cancers

“Among them, three case-control studies showed that 503 daily high dose consumption of green tea was inversely associated with epithelial ovarian 504 cancer occurrence (OR: 0.82, 95% CI [0.38; 1.79]) [105], (OR: 0.43, 95% CI: [0.30; 0.63]) [106], 505 (OR: 0.46, 95% CI [0.26; 0.84]) [107]”

Based on these sentence just in two study there was a relationship. Study with (OR: 0.82, 95% CI [0.38; 1.79]): relationship in cite 105 was not significant, because 95% CI include the number 1. Please revise the above mentioned sentences.

Author’s response: Thank you for pointing that out. The sentences have been revised.

Changes in manuscript are: Several epidemiological studies examined the effect of green tea consumption on ep-ithelial ovarian cancer occurrence. Among them, two case-control studies showed that daily high dose consumption of green tea was inversely associated with epithelial ovarian cancer occurrence  (OR: 0.43, 95% CI: [0.30; 0.63]) [119], (OR: 0.46, 95% CI [0.26; 0.84]) [120]. However, two other studies did not find any association between the two (OR: 0.90, 95% CI [0.50; 1.61]) [121], (OR: 0.82, 95% CI [0.38; 1.79][122] but a meta-analysis did con-firm the presence of a statistically significant inverse relationship (OR: 0.66, 95% CI [0.54; 0.80]) [123].

Please revise the Table 1. Add the references of each study in the table 1.  2. Omit the PMID studies, 3. Include the other characteristics of studies like year, sample size, type of studies, ..

Author’s response: Thank you for this suggestion. Table 1 and Table 2 have become table 3 and Table 4, which were revised. PMID studies have been omitted, references added as well as the year, sample size and type of studies.

Changes in manuscript are:

Study & Year

Type of Study

Cases/Size

Exposure vs Control

OR (95% CI)

Ovarian Cancer

Nagle et al. (2010)

[120]

Case-control

1,271/1,198

Never vs ≥ 1 cup/day

0.82 (0.38-1.79)

Zhang et al. (2002)

[121]

Case-control

254/652

Never or rarely vs ≥ 1 time/day

0.43 (0.30-0.63)

Song et al. (2008)

[122]

Case-control

781/1,263

Never or rarely vs ≥ 1 cup/day

0.46 (0.26-0.84)

Goodman et al. (2003)

[123]

Case-control

164/194

Never vs ≥ 1 cup/day

0.90 (0.50-1.61)

Endometrial Cancer

Gao et al. (2005)

[147]

Case-control

995/1,087

Never vs ≥ 7 cups/day

0.76 (0.60-0.95)

Kakuta et al. (2009)

[148]

Case-control

152/285

4 cups/week vs ≥ 4 cups/day

0.33 (0.15-0.75)

Hirose et al. (2007)

[149]

Case-control

229/2,425

Never vs ≥ 7 cups/day

1.33 (0.75-2.35)

Shimazu et al. (2008)

[150]

Prospective cohort

117/53,724

4 cups/week vs ≥5 cups/day

0.75 (0.44-1.30)

Xu et al. (2007)

[151]

Case-control

1,204/1,212

Never vs ever green tea

0.80 (0.60-0.90)

Table 3. Observational studies examining the effect of green tea on reproductive cancers

Study

Phase

Intervention vs Control

Length of Intervention

Results

Ovarian Cancer

(FIGO stage III-IV serous or endometrioid ovarian cancer)

Trudel et al. [151]

NCT00721890

Phase II clinical trial

N=16

Nonrandomized, single-arm, two-stage design. All treated with double-brewed green tea 500 mL (~639 mg/mL EGCG) for up to 18 months

Primary outcome absence of recurrence at 18 months

Green tea intake for minimum less than 100 days; maximum more than 3 years

Absence of recurrence of ovarian cancer 5/16 women at 18 months

Trial terminated

Cervical Intraepithelial Neoplasia (CIN)

Garcia et al. [152]

NCT00303823

Phase II clinical trial

N=98

50 treated and 48 control

(41 in each group were analysed)

Randomized to Polyphenon E (800mg of EGCG) vs placebo

Primary outcomes: HPV clearance or CIN1 resolution at 4 months

Polyphenon E or placebo intake for 16 weeks

Follow up after 2 weeks of treatment completion

Clearance was similar. Progression of cervical lesion in 14.6% of women receiving intervention vs 7.7% in those receiving placebo

Ahn et al.

[153]

Clinical trial

N=51

27 treated and 39 control

Randomized to Polyphenon E and EGCG ointment or 200 mg capsule vs untreated control group

Primary outcomes: HPV DNA titers, histology or cytology

Polyphenol E ointment or 200 mg capsule or 200 mg EGCG capsule taken for 8 to 12 weeks

69% (35/51 patients)response in group taking green tea extract vs 10% (4/39 patients) response in control group

Vulvar  Usual type of vulvar Intraepithelial neoplasia (uVIN)

Yap et al.

[154]

Phase II clinical trial

N=26

13 treated and 13 control

Randomized 1:1 to Sinecatechins 10% vs placebo ointment

Primary outcome was histological resolution of uVIN at 32 weeks

Sinecatechins ointment or placebo for 16 weeks

Follow up at 2,4,8,16,32 and 52 weeks

5/13 patients who received sinecatechins had complete clinical response and 8/13 had partial response

 Table 4. Interventional trials studying the effect of green tea on reproductive cancers

Reviewer 2 Report

Although the paper delivers a good topic and is well-presented, some important improvements should be performed before further consideration.

  1. I recommend the authors highlight information about the databases used for collecting/extracting the data (for example, Web of Science, Scopus, Google Scholar,..) and what keywords were used during the literature search along with the period of studies included in the review. This ensures that the paper covers all recent and relevant studies. All these points could be highlighted, at least, in the introduction section.
  2. In the Introduction section, I recommend the authors add additional information about the catechins of green tea, including EGCG as promising anticancer agents against reproductive cancers generated by human herpesvirus infections. I recommend the authors use the following references (DOI: 10.3390/ijms24010247) to extract the recommended information.
  3. Also, I recommend the authors discuss the current technologies and future directions of the possibility of improving the phytochemical content of EGCG to produce a natural product with enhanced anticancer properties.
  4. Since the paper focuses on various reproductive cancers targeted by EGCG, the effective concentration or doses should be highlighted. This information can be extracted from the cited references of the reviewed studies. Alternatively, the authors can mention the range of the effective concentrations or doses. This will aid other researchers to have an estimation of concentrations/doses that could be used in further investigations. 
  5. Finally, I recommend the authors double-check the whole manuscript for typing errors. 

Author Response

Response: Manuscript ID cancers-2165764

Dear Editor,

We thank you and the reviewers for their helpful suggestions regarding the manuscript entitled: “Green Tea in Reproductive Cancers: Could Treatment be as Simple?”

We have addressed each of the reviewers’ comments point-by-point in the comments below. In the revised manuscript, altered text is shown by “track changes”. The authors believe the suggestions of the reviewers have greatly improved the manuscript.

We hope these changes will be satisfactory and look forward to any further suggestions.

Best wishes,

Md Soriful Islam,

for the authors

Response to Reviewer 2 comments:

Reviewer 2

Comments and Suggestions for Authors

Although the paper delivers a good topic and is well-presented, some important improvements should be performed before further consideration.

I recommend the authors highlight information about the databases used for collecting/extracting the data (for example, Web of Science, Scopus, Google Scholar,..) and what keywords were used during the literature search along with the period of studies included in the review. This ensures that the paper covers all recent and relevant studies. All these points could be highlighted, at least, in the introduction section.

Author’s response: Thank you for this suggestion. The information about the databases and keywords used for extracting the studies is included in the methods.

Changes in manuscript (methods) are: A comprehensive search of PubMed and Google Scholar up to December 2022 was con-ducted. The following keywords were used: green tea, EGCG, bioavailability, chemical composition, chemical structure, green tea metabolism, pharmacokinetics, pharmacody-namics, EGCG mechanisms, antioxidant, prooxidant, lipid peroxidation, apoptosis, epi-genetic regulations, ovarian cancer, cervical cancer, endometrial cancer, vulvar cancer. The methodology of this study was that of a scoping review. This was conducted to iden-tify current evidence of EGCG as a possible treatment or adjunct agent for gynecologic cancers and to identify and define possible knowledge gaps.

In the Introduction section, I recommend the authors add additional information about the catechins of green tea, including EGCG as promising anticancer agents against reproductive cancers generated by human herpesvirus infections. I recommend the authors use the following references (DOI: 10.3390/ijms24010247) to extract the recommended information.

Author’s response: Thank you for the suggestion. This information was added to the introduction to highlight the fact that there are many potential applications for the use of EGCG as a therapeutic in cancer phenotypes driving from a variety of sources, as well as the idea that there is existing research on EGCG in other cancer types outside of the scope of this paper.

Changes in manuscript (introduction) are: . EGCG has proven benefits in chemoprevention as seen in studies of breast, prostate, kid-ney, colon, and liver cancer [10-13]. EGCG works through multiple processes to inhibit cell invasion and metastasis. It has also been shown that EGCG can mitigate the cancer driv-ing effects of human herpesviruses infections [14]. These infections can drive a plethora of cancer phenotypes, including nasopharyngeal carcinoma, gastric cancer, squamous cell carcinoma, as well as several gynecological cancers subtypes [15-17].

Also, I recommend the authors discuss the current technologies and future directions of the possibility of improving the phytochemical content of EGCG to produce a natural product with enhanced anticancer properties.

Author’s response: Thank you so much for suggesting this. We added this information under the section “Green Tea: Chemistry and Bioavailability” to highlight the use of pro-EGCG analogs in different studies for enhanced chemical properties and bioavailability. We also discussed the improved anticancer effects that have been demonstrated with the use of EGCG prodrugs.

Changes in manuscript are: In attempt to overcome the poor bioavailability of green tea EGCG, several groups have utilized synthetic EGCG derivatives, also known as EGCG prodrugs generated by acetylation of the reactive hydroxyl groups [52]. This was shown to protect the hydroxyl groups and enhance the stability, bioavailability and biological potency of EGCG [53]. Pro-EGCG analogs have demonstrated greater anti-oxidative and anti-angiogenic capaci-ties than EGCG in mice endometrial implants as well as human leiomyoma cells [54,55]. Furthermore, Pro-EGCG exerted significantly stronger antitumor effects than EGCG in skin and breast cancer cell lines and experimental cancer mouse models [56-60]. These findings provide insight into the enhanced properties of Pro-EGCG analogs and their po-tential use for cancer prevention and treatment.

Since the paper focuses on various reproductive cancers targeted by EGCG, the effective concentration or doses should be highlighted. This information can be extracted from the cited references of the reviewed studies. Alternatively, the authors can mention the range of the effective concentrations or doses. This will aid other researchers to have an estimation of concentrations/doses that could be used in further investigations.

Author’s response: Thank you for this great suggestion.  The effective concentration of the green tea extract or EGCG was added in the text as well as in tables 1 and 2.

Finally, I recommend the authors double-check the whole manuscript for typing errors.

Author’s response: Thank you for this recommendation. The whole manuscript was reviewed for typing errors.

Round 2

Reviewer 1 Report

Thank you for the revision

Reviewer 2 Report

The manuscript has been significantly improved.